# A Performance Prediction Model for Pumps as Turbines (PATs)

**Stefania Fontanella [1],\*, Oreste Fecarotta [2] , Bruno Molino [3], Luca Cozzolino [1] and Renata Della Morte [1]**

[1]  Department of Engineering of the University of Naples 'Parthenope', Centro Direzionale Isola C4, 80125 Naples, Italy; luca.cozzolino@uniparthenope.it (L.C.); renata.dellamorte@uniparthenope.it (R.D.M.)

[2]  Department of Civil, Architectural and Environmental Engineering, University of Naples "Federico II", Via Claudio 21, 80125 Naples, Italy; oreste.fecarotta@unina.it

[3]  Department of Bioscience and Territory, University of Molise, Via Francesco De Sanctis, 1, 86100 Campobasso, Italy; bruno.molino@unimol.it

\*  Correspondence: stefania.fontanella@uniparthenope.it

**Abstract:** In recent years, the interest towards the use of pumps operating as turbines (PATs) for the generation of electrical energy has increased, due to the low cost of implementation and maintenance. The main issue that inhibits a wider use of PATs is the lack of corresponding characteristic curves, because manufacturers usually provide only the pump-mode performance characteristics. In the PAT selection phase, the lack of turbine-mode characteristic curves forces users to expend expensive and time-consuming efforts in laboratory testing. In the technical literature, numerous methods are available for the prediction of PAT turbine-mode performance based on the pump-mode characteristics, but these models are usually calibrated making use of few devices. To overcome this limit, a performance database called Redawn is presented and the data collected are used to calibrate novel PAT performance models.

**Keywords:** Pumps as Turbine; energy recovery; performance prediction; best efficiency performances; experimental database; water distribution network; hydropower

## 1. Introduction

Clean energy production for the reduction of greenhouse gas emissions and the mitigation of the global warming treat are becoming increasingly important, and this is prompting attention towards the adoption of alternative energy uses [1–3]. Among these uses, approaches based on energy recovery are increasingly attractive [4,5], and researchers have recently focused their attention on water distribution systems (WDS).

The management of pressure in Water Distribution Systems (WDS) is a strategy commonly implemented to mitigate the issue of leakage because water losses increase with pressure [6]. Of course, pressure reduction valves (PRVs) may be used to reduce the hydraulic head that exceeds the minimum required level for water demand satisfaction, but this strategy seems inefficient in the context of the Water–Energy Nexus [7–9]. Interestingly, industrial pumps can be operated as turbines by inverting the water flow. This implies that Pumps as Turbines (PATs) can be used to convert the head excess into energy, aiming at the double objective of green energy production and reduction of leakage. Of course, the reduction of efficiency with respect to classic turbines is well compensated by the modest installation and maintenance cost, because pumps do not require qualified operators for maintenance [10–12]. For PATs in the range of 1–500 kW, the capital payback period is equal to two years or less, which is shorter than that of the corresponding turbines [10,11]. For these reasons, the use of reverse pumps that

work as turbines (PATs) is becoming a good alternative to PRVs, which dissipate flow head without converting it into electric energy [10–13]. A distinct advantage of PATs is their operating flexibility, because the PAT operating point can be modulated in many ways [14–16], bypassing part of the flow and activating a PRV in parallel with the PAT to dissipate the excess of head drop, or variating the PAT's rotation speed by means of an inverter drive [17–19].

From the perspective of applicability, the critical issue of PAT implementation is the lack of characteristic curves and related performance data, which inhibits the quick selection of an appropriate model based on the manufacturers' catalogs. The lack of information requires expensive and time-consuming efforts in terms of laboratory testing. To overcome this issue, several models that predict PAT performances have been proposed in the literature. These models are generally formulated with reference to the Best Efficiency Point (BEP) operating condition, which is defined as the performance corresponding to the maximum efficiency. For example, Stepanoff [20] defined the relationships among the pump and turbine mode flow rate, head, and hydraulic efficiency as a function of pump efficiency, while Childs [21] proposed the relationships between pump- and turbine mode powers. Hancock modified the equation proposed by Childs by assuming that that BEP efficiency in pump mode was quite similar to BEP efficiency in reverse mode [22]. Grover [23] and Hergt et al. [24] proposed relationships as a function of the PAT specific speed. Using experimental analysis, Alatorre-Frenk and Thomas [25] defined correlations which help to estimate the PAT flow rate and head at BEP as a function of the pump efficiency.

In the last decade, many researchers have focused their attention on the possibility of estimating the characteristic curves of pumps operating as turbines [26]. Derakhshan and Nourbakhsh proposed an approach for predicting the centrifugal PAT performances from the pump specific speed [27]. Similarly, Nautiyal et al. [28] obtained correlations for the horizontal axis single-stage PATs as a function of the pump specific speed and the efficiency at BEP point in pump mode. Yang et al. [29] calibrated their analysis of velocity triangles in direct and reverse mode using the experimental data by Williams [30], Singh and Nestmann [19], Singh [31], and Joshi et al. [32]. They also proposed a correction of the relationship introduced by Derakhshan and Nourbakhsh [27]. Tan and Engeda [33] correlated the BEP hydraulic characteristics in turbine mode with the specific diameter. Moreover, they defined a correlation between the specific speed in pump mode $Ns_p$ and the ratio efficiency in pump and turbine mode. Barbarelli et al. [34] developed an operative procedure for optimal PAT selection composed of four phases tested for six pumps with specific speed between 9 and 65. As an alternative to laboratory experiments, Computational Fluid Dynamics (CDF) methods have been used to forecast PAT performance, trying to overcome the difficulties due to the time-consuming and expensive laboratory activities. Of course, the CFD approach introduces additional difficulties, due to the credibility of the mathematical model and of the numerical approach used [25,35,36].

A major issue with the methods proposed in the literature for the prediction of turbine mode performance is that they are calibrated using a small number of devices, introducing significant errors when the corresponding results are compared with experimental results that are outside of the calibration range. In the present paper, an increased database of pump and turbine performance data, collected in the context of the REDAWN project, is presented. This database is used to calibrate new relationships between pump and turbine mode, showing that the BEP pump and turbine mode conditions are mainly correlated by the rotational speeds in pump and turbine mode, and supplying novel characteristic curves in turbine mode. The approach proposed allows quick and easy estimation of the turbine performance for numerous pump models and flow conditions, improving existing approaches.

## 2. Data Available

The REDAWN (Reduction Energy Dependency in Atlantic area Water Networks) project has made available a database, called Redawn, that contains the main geometric and performance characteristics of 34 different centrifugal pump models. These data were extracted from existing literature [19,25,34,37] or supplied by manufacturers and participating researchers. Concerning the device, four different

types are available, as follows [38]: 20 ESOB (End Suction Own Bearing) devices, 7 MSV (Multi-Stage Vertical), 6 MSO (Multi-Stage Horizontal), and one MSS (Multi-Stage Submersible).

For each pump, the database contains the following data: manufacturer, pump model, pump type, diameter, number of stages, specific speed in turbine condition, the characteristic curve for the pump in reverse and direct mode, and the BEP hydraulic characteristics in pump- and turbine-mode. For some devices, different turbine-mode rotation speeds are available, thus resulting in 52 different turbine-mode devices. In Table 1, the main characteristics of the database are resumed as follows (from left to right): device code, manufacturer, type of pump, and number K of different rotational speeds in turbine mode considered for the device.

**Table 1.** Resume of Redawn database characteristics.

| Device Code | Manufacturer | Type | K |
|---|---|---|---|
| 'Etanorm 32-125' | KSB (Frankenthal, Germany) | ESOB | 1 |
| 'Etanorm 50-160' | KSB (Frankenthal, Germany) | ESOB | 1 |
| 'FHE80-200' | Lowara (Vicenza, Italy) | ESOB | 2 |
| 'Etanorm 150-200' | KSB (Frankenthal, Germany) | ESOB | 1 |
| 'Etanorm 100-315' | KSB (Frankenthal, Germany) | ESOB | 1 |
| 'Etanorm 50-315?' | KSB (Frankenthal, Germany) | ESOB | 1 |
| 'Etanorm 65-125' | KSB (Frankenthal, Germany) | ESOB | 1 |
| 'Etanorm 65-160' | KSB (Frankenthal, Germany) | ESOB | 1 |
| 'Etanorm 65-200' | KSB (Frankenthal, Germany) | ESOB | 1 |
| 'Etanorm 65-250' | KSB (Frankenthal, Germany) | ESOB | 1 |
| 'Etanorm 65-315' | KSB (Frankenthal, Germany) | ESOB | 1 |
| 'Etanorm 80-200' | KSB (Frankenthal, Germany) | ESOB | 1 |
| 'Etanorm 80-250' | KSB (Frankenthal, Germany) | ESOB | 1 |
| 'Etanorm 80-315' | KSB (Frankenthal, Germany) | ESOB | 1 |
| 'Etanorm 80-400' | KSB (Frankenthal, Germany) | ESOB | 1 |
| 'Etanorm 100-200' | KSB (Frankenthal, Germany) | ESOB | 1 |
| 'Etanorm 100-315' | KSB (Frankenthal, Germany) | ESOB | 1 |
| 'Etanorm 100-400' | KSB (Frankenthal, Germany) | ESOB | 1 |
| 'Etanorm 125-400' | KSB (Frankenthal, Germany) | ESOB | 1 |
| 'Etanorm 150-250' | KSB (Frankenthal, Germany) | ESOB | 1 |
| 'P(E18S64)/1A' | Caprari (Modena, Italy) | MSS | 3 |
| 'P14C/1G' | Caprari (Modena, Italy) | MSV | 3 |
| 'P14C/1A' | Caprari (Modena, Italy) | MSV | 3 |
| 'P14C/1C' | Caprari (Modena, Italy) | MSV | 3 |
| 'P16D/1B' | Caprari (Modena, Italy) | MSV | 3 |
| 'P16C/1A' | Caprari (Modena, Italy) | MSV | 3 |
| 'P18C/1A' | Caprari (Modena, Italy) | MSV | 2 |
| '92SV2G150T_IE3' | Lowara (Vicenza, Italy) | MSV | 4 |
| 'PM50/3' | Caprari (Modena, Italy) | MSO | 1 |
| 'PM50/4' | Caprari (Modena, Italy) | MSO | 1 |
| 'HMU40-2/2' | Caprari (Modena, Italy) | MSO | 1 |
| 'HMU50-1/2' | Caprari (Modena, Italy) | MSO | 1 |
| 'HMU50-2/2' | Caprari (Modena, Italy) | MSO | 1 |
| 'MEC-MR80-3/2A' | Caprari (Modena, Italy) | MSO | 2 |

## 3. Performance Prediction of a PAT

In the present section, the Redawn database is investigated, with the aim of stating which are the parameters of the pumps that influence the reverse condition, defining a new prediction model of the PAT performance for centrifugal pumps.

### 3.1. Specific Speed

The specific speed Ns is a parameter that combines performance and cinematic features of a device (pump or turbine). Among the different definitions of Ns available in the literature [39], the expression

$$Ns = \frac{N \sqrt{Q_b}}{\sqrt[4]{H_b^3}} \tag{1}$$

is used in the following. In Equation (1), $H_b$ (m) and $Q_b$ (m³/s) are the head and the discharge at the BEP point, while N(rpm) is the rotational speed. For the Redawn database, the pump mode specific number $NS_p$ ranges between 6 and 80, while the turbine mode specific number $NS_t$ ranges between 5 and 86.

The specific speed data from the Redawn database, together with those by Chapallaz et al. [40] and Yang et al. [29], are used to find a relationship between $NS_t$ and $NS_p$ (see Figure 1). This relationship is approximately linear, and is expressed as

$$NS_t = 0.8793 \, NS_p \tag{2}$$

which is quite similar to the expression by Chapallaz et al. [40]. It is evident that the turbine mode specific speed is slightly smaller than the pump mode specific speed.

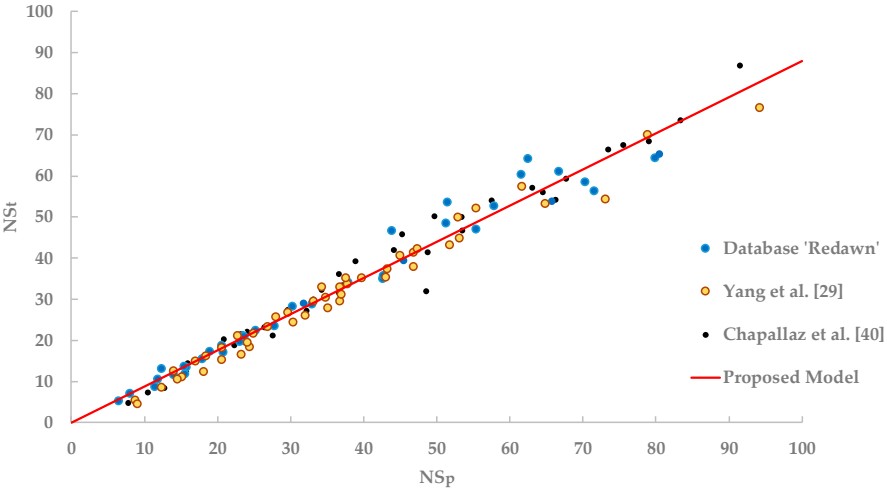

**Figure 1.** Comparison of all available experimental points in the literature and the experimental data of database Redawn.

### 3.2. BEP Performance

Classic similarity laws for pumps and turbines state that the discharge is proportional to the rotational speed, while the head is proportional to the squared rotational speed. With reference to the Redawn database, the relationship between the ratio $Q_{tb}/Q_{pb}$ and the ratio $N_t/N_p$ is shown in Figure 2, where $Q_{tb}$ (L/s) and $Q_{pb}$ (L/s) are the bQBEP turbine mode and pump mode discharges, respectively, while $N_t$ (rpm) and $N_p$ (rpm) are the corresponding rotational speeds. It is evident that this relationship is linear with good approximation, supplying

$$\frac{Q_{tb}}{Q_{pb}} = 1.3595 \frac{N_t}{N_p} \tag{3}$$

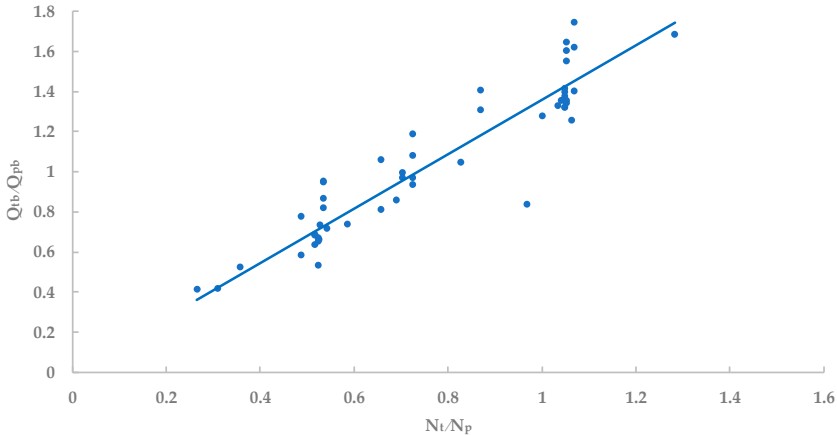

**Figure 2.** Correlation between $\frac{Q_{tb}}{Q_{pb}}$ and $\frac{N_t}{N_p}$.

Equation (3) is valid in the range $0.2658 < N_t/N_p < 1.2828$, given the data available from the Redawn database. Expectedly, the ratio between the BEP discharges in pump and reverse modes mainly depends on the motor features and on the presence of an inverter drive.

The similarity laws predict the dependency of the head on the squared rotational speed. Actually, the substitution of Equation (3) in Equation (2) leads to the simple quadratic relationship

$$\frac{H_{tb}}{H_{pb}} = 1.4568\left(\frac{N_t}{N_p}\right)^2 \tag{4}$$

where $H_{tb}$ (m) and $H_{pb}$ (m) are the bQBEP turbine- and pump-mode heads, respectively. Equation (4), which is valid in the same range of Equation (3), is represented in Figure 3, where the experimental data are also reported. The good agreement between Equation (4) and the experimental data confirms that the BEP hydraulic characteristics in turbine-mode are strongly dependent on the motor features.

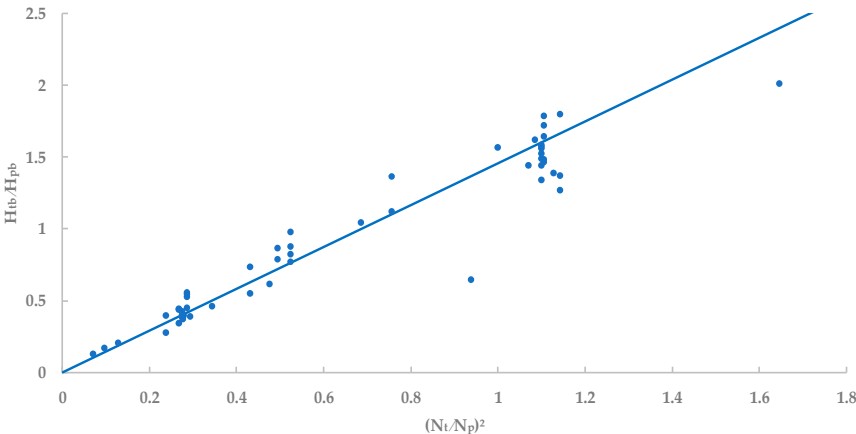

**Figure 3.** Correlation between $\frac{H_{tb}}{H_{pb}}$ and the square of $\frac{N_t}{N_p}$.

The similarity laws for pumps and turbines predict that the power is proportional to the third power of the rotational speed. For this reason, it is convenient to consider the dependency of $P_{tb}/P_{pb}$ on the cube of the ratio $N_t/N_p$. This relationship is elucidated in Figure 4, and the interpolation supplies

$$\frac{P_{tb}}{P_{pb}} = 1.0403\left(\frac{N_t}{N_p}\right)^3 \tag{5}$$

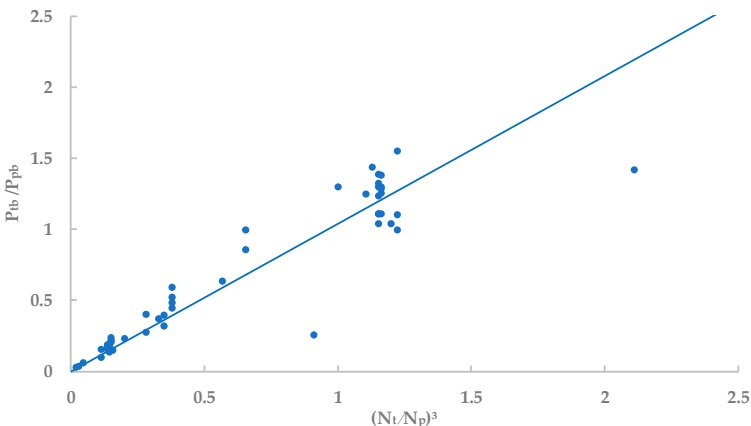

**Figure 4.** Correlation between $P_{tb}/P_{pb}$ and $N_t/N_p$.

Equation (5) is valid in the same range of Equations (3) and (4). Recalling the definition

$$\eta_{tb} = \frac{P_{tb}}{H_{tb}Q_{tb}} \tag{6}$$

Equations (3)–(5) can be used to evaluate the turbine-mode BEP efficiency $\eta_{tb}$.

### 3.3. Characteristic Curves

The turbine mode characteristic curves state the relationship between the turbine mode power $P_t$, head $H_t$, and discharge $Q_t$ for a given rotational speed $N_t$ and for functioning conditions different from the BEP. Of course, the dependence on $N_t$ is conveniently eliminated by considering the dependence of the dimensionless variables $P_t/P_{tb}$ and $H_t/H_{tb}$ on $Q_t/Q_{tb}$. For this reason, the database Redawn is investigated in order to find suitable turbine mode characteristic curves in dimensionless form. Interestingly, the MSS device available in the database has a behavior significantly different from that of the other devices, and must be treated separately.

In Figure 5, upper panel, the ($Q_t/Q_{tb}$, $H_t/H_{tb}$) experimental points are reported with blue dots for the ESOB, MSO, and MSV pumps, while the MSS data are plotted with red dots. The inspection of the panel shows that ESOB, MSO, and MSV data are nicely aligned without regard to the rotational speed $N_t$, while the MSS data constitute a separate family. The same can be observed in Figure 5, lower panel, where the ($P_t/P_{tb}$, $H_t/H_{tb}$) experimental points are plotted.

The head-discharge and power-discharge data for the ESOB-MSO-MSV family can be interpolated by means of the following models.

$$\frac{H_t}{H_{tb}} = 1 + 0.9633 \left(\frac{Q_t}{Q_{tb}} - 1\right)^2 + 1.4965 \left(\frac{Q_t}{Q_{tb}} - 1\right)^\infty, \text{ for } 0.33 < \frac{Q_t}{Q_{tb}} < 6.25 \tag{7}$$

$$\begin{aligned}\frac{P_t}{P_{tb}} &= 1 + 2.7071 \left(\frac{Q_t}{Q_{tb}} - 1\right) + 1.4326\left(\frac{Q_t}{Q_{tb}} - 1\right)^2 - 0.2405\left(\frac{Q_t}{Q_{tb}} - 1\right)^3 \\ &+ 0.03499 \left(\frac{Q_t}{Q_{tb}} - 1\right)^4, \text{ for } 0.33 < \frac{Q_t}{Q_{tb}} < 6.25\end{aligned} \tag{8}$$

The MSS head-discharge and power-discharge data can be interpolated by means of the following models

$$\frac{H_t}{H_{tb}} = 1 + 1.2696 \left(\frac{Q_t}{Q_{tb}} - 1\right)^2 + 1.8665 \left(\frac{Q_t}{Q_{tb}} - 1\right)^\infty, \text{ for } 0.47 < \frac{Q_t}{Q_{tb}} < 2.91 \tag{9}$$

$$\begin{aligned}\frac{P_t}{P_{tb}} &= 1 + 2.7169 \left(\frac{Q_t}{Q_{tb}} - 1\right) + 1.9992\left(\frac{Q_t}{Q_{tb}} - 1\right)^2 + 0.1926\left(\frac{Q_t}{Q_{tb}} - 1\right)^3 \\ &- 0.08964 \left(\frac{Q_t}{Q_{tb}} - 1\right)^4, \text{ for } 0.47 < \frac{Q_t}{Q_{tb}} < 2.91\end{aligned} \tag{10}$$

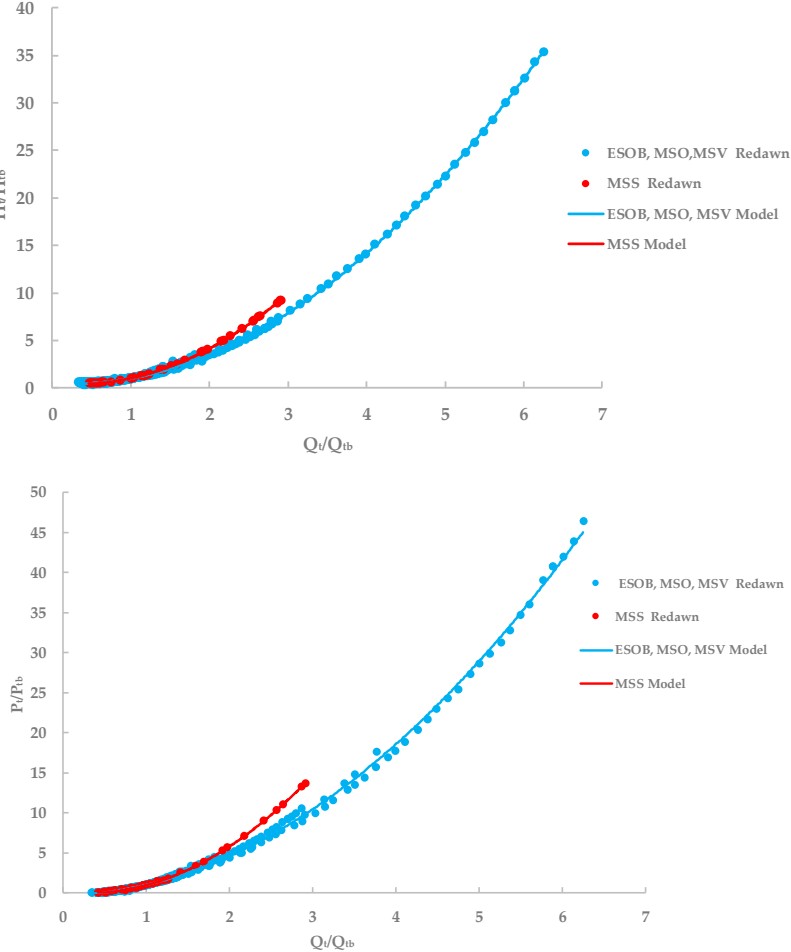

**Figure 5.** Redawn database: head-discharge (**upper** panel) and power-discharge (**lower** panel) dimensionless characteristic curves.

Interestingly, the form of Equations (7)–(10) ensures that $H_t/H_{tb} = 1$ when $Q_t/Q_{tb} = 1$, and that $P_t/P_{tb} \approx 0$ when $Q_t/Q_{tb} = 0$, while $P_t/P_{tb} = 1$ when $Q_t/Q_{tb} = 1$. The corresponding efficiency curves can be obtained from Equations (7) and (8) (ESOB, MSO, and MSV pumps) or from Equations (9) and (10) using the definition

$$\frac{\eta_t}{\eta_{tb}} = \frac{P_t H_{tb} Q_{tb}}{P_{tb} H_t Q_t} \tag{11}$$

In Figure 6, Equation (11) is represented separately for the ESOB-MSO-MSV group and the MSS pump, showing that the congruency condition $\eta_t/\eta_{tb} = 1$ for $Q_t/Q_{tb} = 1$ is nicely satisfied.

In Figure 7, the same curves are compared with efficiency experimental data. It can be observed that the efficiency curve for the ESOB-MSO-MSV pumps nicely interpolates the experimental data, while there is some discrepancy between the experimental data and mathematical model in the case of the MSS pump. The discrepancy between the efficiency MSS interpolated curve and the experimental values is mainly attributed to the paucity of the corresponding data.

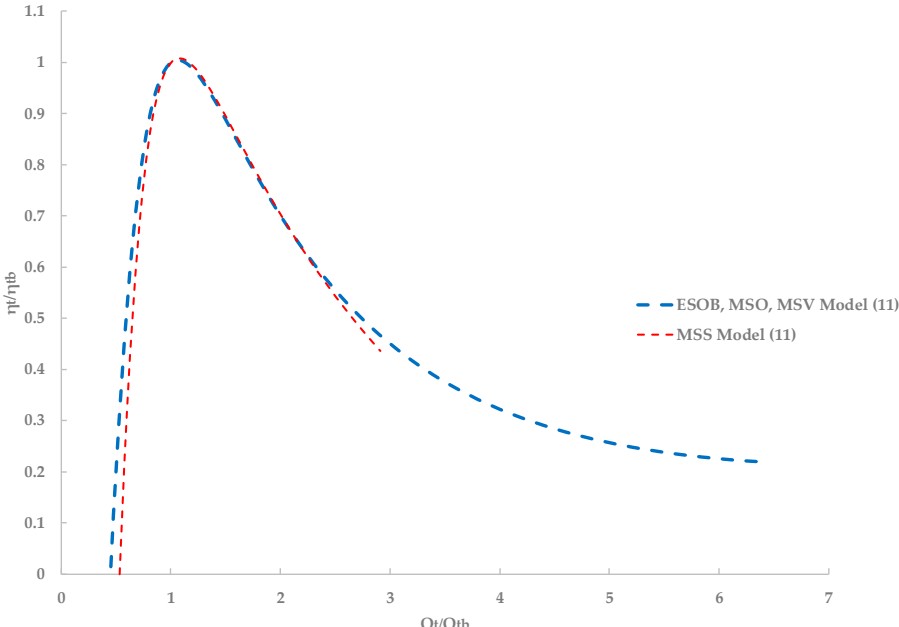

**Figure 6.** Comparison of efficiency curves evaluated using Relationship (11) for the ESOB, MSV, and MSO model with the MSS ones.

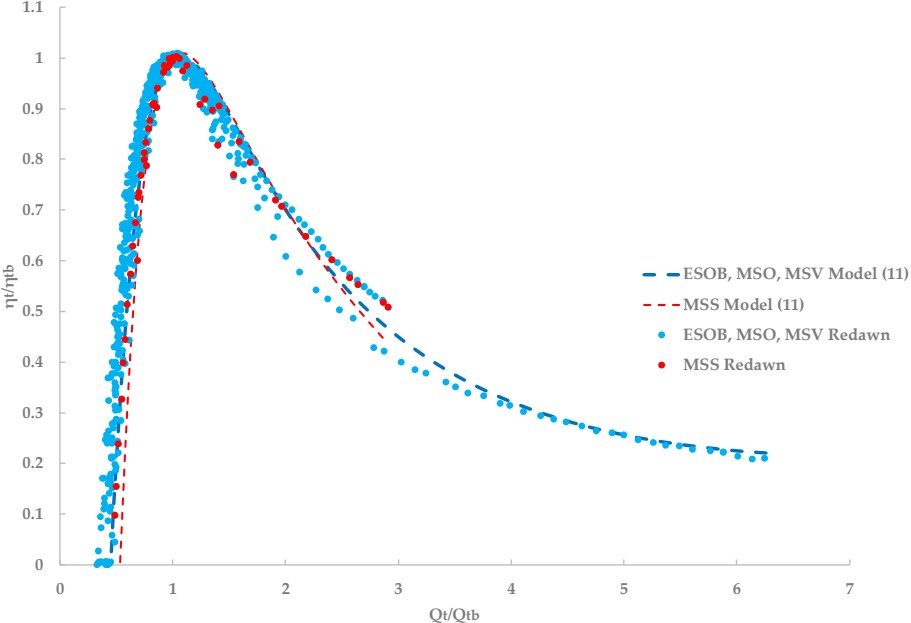

**Figure 7.** Comparison of Redawn database (ESOB, MSO, MSV, MSS) with efficiency-discharge evaluated using Relationship (11).

### 3.4. Comparison with Methods Available in the Literature

It is interesting to compare the average errors exhibited by the models proposed in the present paper with the errors exhibited by the Derakhshan and Nourbakhsh [27] and Tan et Engeda [33] models when they are applied to predict the experimental values contained into the Redawn database. In the present case, the percentage average errors ($E_{AV}$) are defined as follows

$$E_{AV} = \frac{100}{Nd} \sum_{i=1}^{Nd} \left( \frac{Yi_{EX} - Yi_{F}}{Yi_{EX}} \right) \tag{12}$$

where $Yi_{EX}$ is the i-th measured physical quantity, while $Yi_{F}$ is the corresponding computed quantity.

In Table 2, the average errors for the BEP characteristics are reported. The inspection of the table demonstrates the minor ability of the models by Derakhshan and Nourbakhsh [27] and Tan and Engeda [33] to predict the BEP turbine mode for the Redawn data.

**Table 2.** Comparison of the average BEP prediction errors for different models from the literature.

|  | $Q_{tb}$ | $H_{tb}$ | $P_{tb}$ | $\eta_{tb}$ |
|---|---|---|---|---|
| Proposed model | 0.48% | 1.03% | 2.00% | 4.48% |
| Derakhshan & Nourbakhsh [27] | 1.6% | 20.0% | 30.0% | 4.5% |
| Tan et Engeda [33] | 1.2% | 38.0% | 86.0% | 14.0% |

This discrepancy between Redawn data and the models from the literature may be attributed not only to the characteristics of the pumps used, but also to the range of experimental values and even to structural incongruences of the models. To elucidate the last observation, the models proposed by Derakhshan and Nourbakhsh [27] for the prediction of the turbine mode characteristic curves are considered:

$$\frac{Ht}{Htb} = 1.0283 \left(\frac{Qt}{Qtb}\right)^2 - 0.5468 \frac{Qt}{Qtb}^{\infty} + 0.5314 \tag{13}$$

$$\frac{Pt}{Ptb} = -0.3092\left(\frac{Qt}{Qtb}\right)^3 + 2.1472 \left(\frac{Qt}{Qtb}\right)^2 - 0.8865\frac{Qt}{Qtb} + 0.0452 \tag{14}$$

These models are compared in Figure 8 with the Redawn experimental data. The inspection of Figure 8 (upper panel) shows that the head-discharge model by Derakhshan and Nourbakhsh [27] nicely predicts the Redawn data (ESOB, MSO, MSV) for $Q_t/Q_{tb} < 3$, but departs from the experimental data for higher values of the discharge, which seems to highlight the limited range of flow rates considered.

This is confirmed by considering the power-discharge model by Derakhshan and Nourbakhsh [27] (Figure 8, central panel), which departs from Redawn data for $Q_t/Q_{tb} > 2$. Actually, Equation (14) by Derakhshan and Nourbakhsh [27] exhibits a maximum around $Q_t/Q_{tb} = 4.5$, implying that the power predicted in turbine-mode decreases for $Q_t/Q_{tb} > 4.5$, which is unphysical and not confirmed by experimental data. The decreasing behavior is immediately understood considering that the power-discharge model by Derakhshan and Nourbakhsh [27] exhibits a negative coefficient that is multiplied by the cube of $Q_t/Q_{tb}$, producing a concave plot for higher values of $Q_t/Q_{tb}$. An additional minor incongruence is evident, that is the value 0.0452 of the intercept in the power-discharge Derakhshan and Nourbakhsh [27] model, implying that power is produced also for null discharge.

The inspection of Figure 8, lower panel, where the efficiency curve deduced by Derakhshan and Nourbakhsh [27] is represented, shows that the efficiency is heavily underestimated for $Q_t/Q_{tb} > 2$. In particular, the right tail of the experimental data is not captured.

## 4. Application

The PAT characteristic curves were predicted using the model defined above for four different pumps (see BEP characteristics in pump mode in Table 3, while the experimental BEP characteristics in turbine mode are reported in Table 4). In Table 5, the values of the BEP characteristics in turbine mode calculated on the basis of Equations (3)–(6) are reported, together with the corresponding percentage relative errors. In Table 6, the turbine mode BEP characteristics calculated with the Derakhshan and Nourbakhsh [27] model are summarized with the corresponding relative percentage errors. The comparison of Tables 5 and 6 confirms that the novel model offers improved turbine mode BEP evaluations for the range of pumps used.

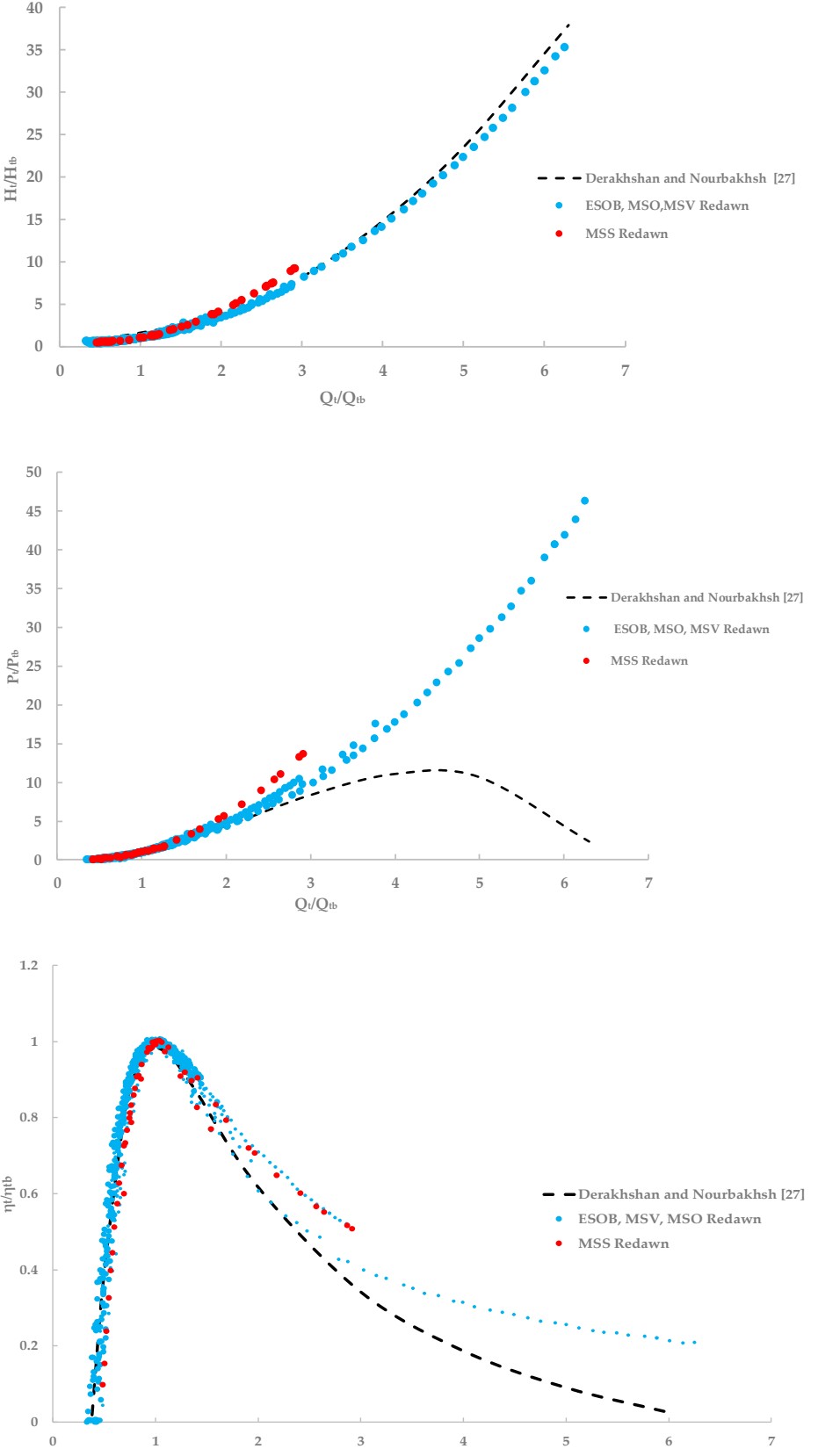

**Figure 8.** Comparison between the Redawn experimental data and the Derakhshan and Nourbakhsh [27] model in terms of; head-discharge (**upper** panel), power-discharge (**central** panel) and efficiency-discharge (**lower** panel).

**Table 3.** Experimental BEP characteristics in pump mode.

| Devices | Manufacturer | $Q_{pb}$ (m$^3$/s) | $H_{pb}$ (m) | $P_{pb}$ (KW) | $\eta_{pb}$ | $N_p$ (rpm) |
|---|---|---|---|---|---|---|
| ESOB, Etanorm 100-400 | KSB (Frankenthal, Germany) | 0.052673 | 49.37302837 | 33.95912663 | 0.750954 | 1450 |
| MSO, MEC-MR80-3/2A | Caprari (Modena, Italy) | 0.042037 | 130.9518891 | 69.89042498 | 0.772358 | 2900 |
| MSV, 92SV2G150T_IE3 | Lowara (Vicenza, Italy) | 0.025474 | 42.28917636 | 13.42392097 | 0.786942 | 2900 |
| MSS, 'P(E18S64)/1A' | Caprari (Modena, Italy) | 0.1964461 | 48.9573971 | 114.3579978 | 0.8246829 | 2935 |

**Table 4.** Experimental BEP characteristics in turbine mode.

| Device | Manufacturer | $Q_{tb}$ (m$^3$/s) | $H_{tb}$ (m) | $P_{tb}$ (KW) | $\eta_{tb}$ | $N_t$ (rpm) |
|---|---|---|---|---|---|---|
| ESOB, Etanorm 100-400 | KSB (Frankenthal, Germany) | 0.072615 | 77.57348 | 41.93998986 | 0.759266 | 1520 |
| MSO, MEC-MR80-3/2A | Caprari (Modena, Italy) | 0.030197 | 51.0721 | 10.41159796 | 0.68847 | 1570 |
| MSV, 92SV2G150T_IE3 | Lowara (Vicenza, Italy) | 0.026722 | 44.25196 | 8.521971903 | 0.734943 | 2400 |
| MSS, 'P(E18S64)/1A' | Caprari (Modena, Italy) | 0.1447000 | 19.5269 | 18.7352210 | 0.6761843 | 1550 |

**Table 5.** PAT performance at BEP, evaluated using the proposed model.

| Devices | $Qtb_{EV}$ (m$^3$/s) | $E_{Qtb}$ | $Htb_{EV}$ (m) | $E_{Htb}$ | $Ptb_{EV}$ (KW) | $E_{Pt}$ | $\eta t_{EV}$ | $E_{\eta t}$ |
|---|---|---|---|---|---|---|---|---|
| ESOB, Etanorm 100-400 | 0.0750659 | −3.37% | 79.03889 | −1.89% | 40.6951 | 2.97% | 0.6992 | 7.91% |
| MSO, MEC-MR80-3/2A | 0.0309395 | −2.46% | 55.91328 | −9.48% | 11.5367 | −10.81% | 0.6798 | 1.26% |
| MSV, 92SV2G150T_IE3 | 0.0286611 | −7.26% | 42.19448 | 4.65% | 7.9155 | 7.12% | 0.6672 | 9.22% |
| MSS, 'P(E18S64)/1A' | 0.1410412 | 2.53% | 19.89140 | −1.87% | 17.5225 | 6.47% | 0.6367 | 5.84% |

**Table 6.** PAT performance at BEP, evaluated using Derakhshan and Nourbakhsh [27].

| Devices | $Qtb_D$ (m$^3$/s) | $E_{Qtb}$ | $Htb_D$ (m) | $E_{Htb}$ | $Ptb_D$ (KW) | $E_{Pt}$ | $\eta t_D$ | $E_{\eta t}$ |
|---|---|---|---|---|---|---|---|---|
| ESOB, Etanorm 100-400 | 0.083529 | −15.03% | 104.217 | −34.35% | 53.33965 | −27.18% | 0.62486 | 17.70% |
| MSO, MEC-MR80-3/2A | 0.0331197 | −9.68% | 75.92709 | −48.67% | 12.72625 | −22.23% | 0.51609 | 25.04% |
| MSV, 92SV2G150T_IE3 | 0.0319179 | −19.45% | 49.65282 | −12.20% | 11.65847 | −36.80% | 0.75019 | −2.07% |
| MSS, 'P(E18S64)/1A' | 0.099137 | 31.49% | 15.3896 | 21.19% | 13.134578 | 29.89% | 0.87793 | −29.84% |

In Figure 9, the experimental characteristic curves for the MSV device 92SV2G150T_IE3 are represented. In the upper panel of the figure, the turbine mode head values for different values of the discharge are compared with those calculated by means of Equation (7), where the BEP characteristics are estimated with Equations (3) and (4). In the same panel, the values calculated by means of the Derakhshan and Nourbakhsh [27] model are represented, showing a less satisfactory agreement between model and experimental data.

In the central panel of Figure 9, the turbine mode power for different values of the flow rate are compared with those calculated by means of Equation (8), where the BEP characteristics are estimated using Equations (3) and (5). In the same panel, the values calculated by means of the Derakhshan and Nourbakhsh [27] model are represented, apparently showing a good agreement with experimental data. This impression is not confirmed by the lower panel of Figure 9, where the efficiency values are represented. In particular, this panel shows that the model by Derakhshan and Nourbakhsh [27] fails when evaluating the BEP characteristics, and this leads to a shift of the efficiency curve.

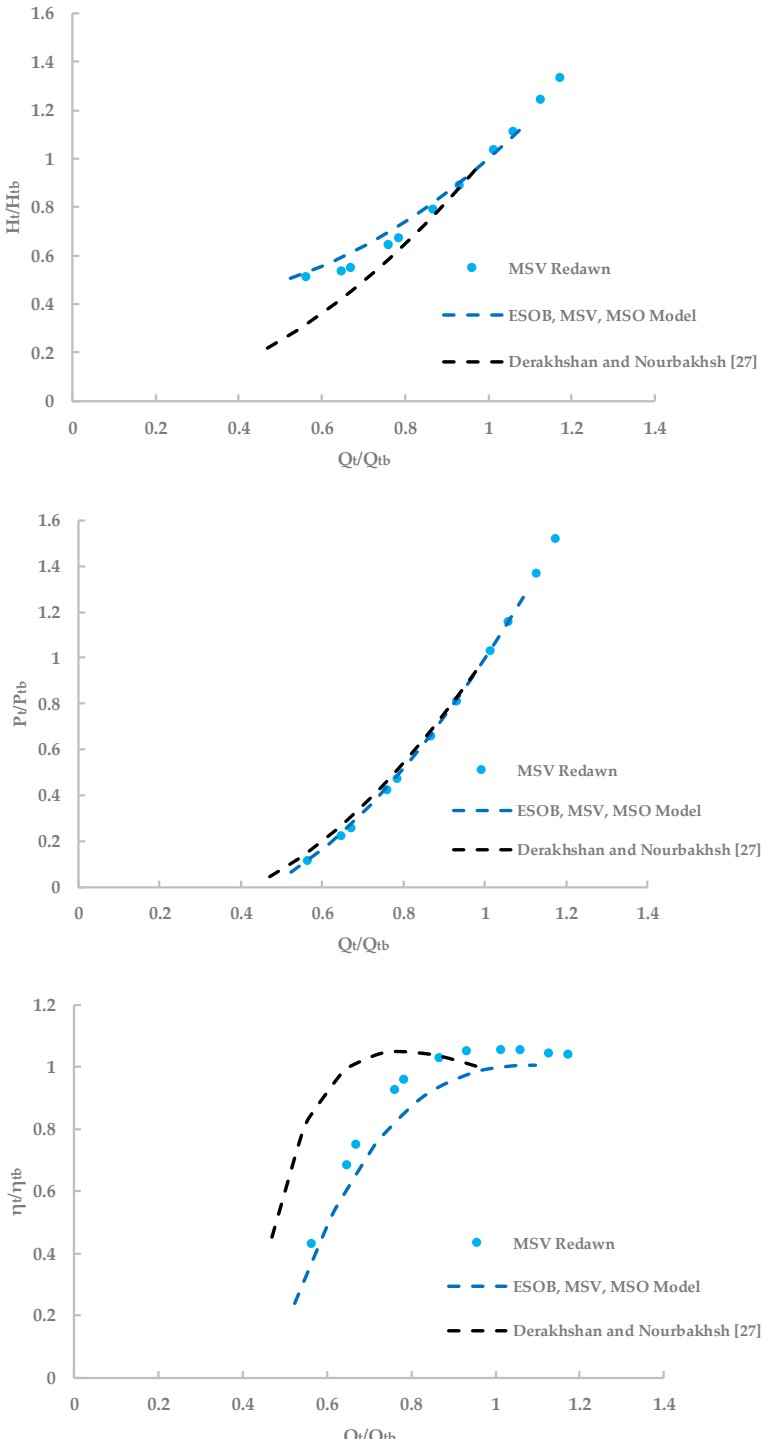

**Figure 9.** Characteristic curves for the 92SV2G150T_IE3 MSV pump. Experimental data (dots), proposed model (blue dashed line), and Derakhshan and Nourbakhsh [27] model (black dashed line): head-discharge (**upper** panel), power-discharge (**central** panel), and efficiency-discharge (**lower** panel).

The exercise is repeated in Figure 10 for the MSS 'P(E18S64)/1A' device using Equations (3)–(6), (9), and (10). In this case, the comparison shows that the Derakhshan and Nourbakhsh [27] produces a shift of all the characteristic curves, due to the errors introduced in the calculation of the BEP values.

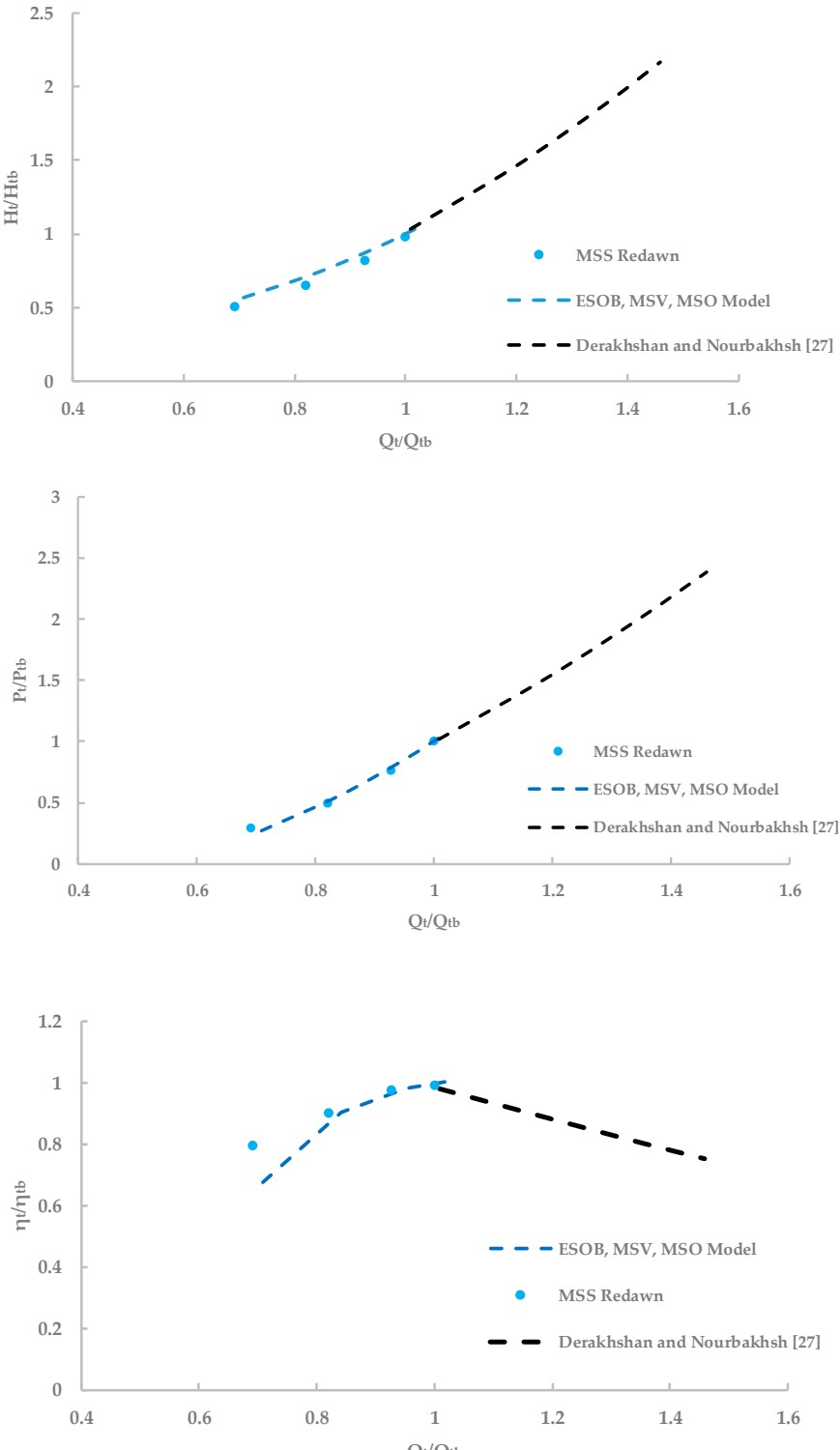

**Figure 10.** Characteristic curves for the 'P(E18S64)/1A' MSS pump. Experimental data (dots), proposed model (blue dashed line), and Derakhshan and Nourbakhsh [27] model (black dashed line): head-discharge (**upper** panel), power-discharge (**central** panel), and efficiency-discharge (**lower** panel).

## 5. Conclusions

The use of pumps operated as turbines (PATs) is arousing increasing interest, due to the reduced costs in comparison with classic turbines. Nonetheless, a major obstacle inhibits the practical application of PATs in actual projects, namely the lack of performance data (best efficiency point data, characteristic

curves). The availability of relationships between the pump and turbine mode performance data could remove this obstacle, at least in preliminary design stages.

The Redawn database, which was produced in the context of the REDAWN project, collects the performance data of 34 pumps operated as turbines at different rotational speeds, resulting in a total of 52 turbine mode devices. The experimental data contained in Redawn were used to produce models able to predict the PATs' performance for a wide range of discharges, heads, and rotational speeds.

The inspection of the experimental data shows that the turbine mode BEP characteristics (head, discharge, power) are related to the corresponding pump mode BEP characteristic through the turbine and pump rotational speeds. This result is familiar, since it descends from the classic similarity laws for pumps and turbines. Conversely, the turbine mode BEP efficiency is a fixed fraction of the pump mode BEP efficiency.

Not only the BEP data, but also the characteristic curves are necessary for preliminary design. When the characteristic curves data obtained from the family of ESOB (End Suction Own Bearing), MSV (Multi-Stage Vertical), and MSO (Multi-Stage Horizontal) pumps, are nondimensionalized with respect to the corresponding turbine mode BEP data, they tend to superpose. This allows obtaining the corresponding dimensionless head-discharge, power-discharge, and efficiency-discharge curves, which do not depend on the rotational speed (the BEP data do, of course, depend on the rotational speeds). Interestingly, the MSS (Multi-Stage Submersible) data follow a different behavior, and appropriate models were calibrated.

With reference to the Redawn dataset, the new models were compared with models available from the literature (Derakhshan and Nourbakhsh [27] and Tan and Engeda [33]), revealing not only some incongruence of existing formulations, but also demonstrating the reliability increase of the novel models.

**Author Contributions:** Conceptualization, S.F.; methodology, S.F. and L.C.; software, S.F.; validation, S.F.; formal analysis, S.F.; investigation, S.F.; resources, S.F. and O.F.; data curation, S.F. and O.F.; writing—original draft preparation, S.F.; writing—review and editing, L.C. and R.D.M.; visualization, S.F., O.F, B.M., L.C. and R.D.M.; supervision, L.C. and R.D.M.; project administration, L.C. and R.D.M.; funding acquisition, O.F. All authors have read and agreed to the published version of the manuscript.

**Funding:** This work was funded through the University of Naples Federico II by the European program "ERDF (European Regional Development Fund) Interreg Atlantic Area Program 2014–2020", through the REDAWN project (Reduction Energy Dependency in Atlantic area Water Networks)—EAPA 198/2016; and the data were made available in this work in a scientific cooperation between the Department of Civil, Architectural and Environmental Engineering of the University of Naples Federico II and the Department of Engineering of the University of Naples 'Parthenope'.

**Conflicts of Interest:** The authors declare no conflict of interest.

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
