# Peer review of "A Performance Prediction Model for Pumps as Turbines (PATs)"

_water, doi:10.3390/w12041175_

Round 1
Reviewer 1 Report
see attachment

Reviewer 2 Report
The article provides a novel prediction model of PAT performance characteristics (best efficiency point, characteristic curves) obtained by post-processing of data collected in a new richer database. As also stated by the authors, these results are useful at least in preliminary design stages of a new energy recovery system based on PAT technology.
The article is generally well written. The text could be rechecked once again in order to remove the eventual remaining grammar errors (preferable by an English native speaker).
Complementary remarks:
Line 21: “this obliges the users to” instead of “this obliges the users obliges”.
Lines 46: “shorter than that of corresponding” instead of “shorter than that corresponding”.
Lines 50-51: this part of phrase must be completed “for example bypassing part of the 50 flow rate if the PAT’s…”.
Line 155: the word “point” is not necessary anymore (already contained in “BEP”).
Line 157: “reverse-mode” instead of “reverse-node”.
Section 4.3: the authors are asked to specify how they defined the errors to compare the different models.
Redawn database: is this database public? Could it be later completed by other academics (or even pump manufacturers) with available data from their specific projects in order to constantly improve the prediction models?
Reviewer 3 Report
The paper present interesting and important topic. The missing of real data and experience by PATs brought authors to try with statistical methods and the databank preparation.
The publication clear and transparently presents the results.
Two general remarks:
- Will be nice to see a comparison with standard Francis turbines as pumps.
- The paper is concentrated on BEP (Best efficiency point) and will be nice to see how the pumps perform in other turbine modes. Could same correlations been used in such cases?
Round 2
Reviewer 1 Report
Basically, the authors made major changes to the revised manuscript, taking into account my suggestions and comments. Now I can recommend the current manuscript for publication in the Water journal.